# Elevated neuron-specific enolase level is associated with postoperative delirium and detection of phosphorylated neurofilament heavy subunit: A prospective observational study

**Kazuhito Mietani**[1], **Maiko Hasegawa-Moriyama**[2], **Reo Inoue**[1], **Toru Ogata**[3], **Nobutake Shimojo**[4], **Makoto Kurano**[5], **Masahiko Sumitani**[6]*, **Kanji Uchida**[1]

1 Department of Anaesthesiology and Pain relief Center, The University of Tokyo Hospital, Tokyo, Japan, 2 Department of Pain and Palliative Medical Sciences, Faculty of Medicine, The University of Tokyo, Tokyo, Japan, 3 Department of Rehabilitation Medicine, The University of Tokyo Hospital, Tokyo, Japan, 4 Department of Emergency and Critical Care Medicine, Tsukuba University Hospital, Ibaraki, Japan, 5 Department of Clinical Laboratory Medicine, The University of Tokyo Hospital, Tokyo, Japan, 6 Department of Pain and Palliative Medicines, Faculty of Medicine, The University of Tokyo, Tokyo, Japan

* sumitanim-ane@h.u-tokyo.ac.jp

**Data Availability Statement:** All relevant data are within the supporting information.

## Abstract

### Background

Delirium is the most common central nervous system complication after surgery. Detection of phosphorylated neurofilament heavy subunit in the serum reflects axonal damage within the central cervous system and is associated with the severity of postoperative delirium. Neuron-specific enolase and S100 calcium-binding protein β have been identified as possible serum biomarkers of postoperative delirium. This study examined the association of the levels of these markers with incidence of postoperative delirium and detection of phosphorylated neurofilament heavy subunit.

### Methods

This study represents a post hoc analysis of 117 patients who participated in a prospective observational study of postoperative delirium in patients undergoing cancer surgery. Patients were clinically assessed for development of postoperative delirium within the first five days of surgery. Serum levels of phosphorylated neurofilament heavy subunit, neuron-specific enolase, and S100 calcium-binding protein β levels were measured on postoperative day 3.

### Results

Forty-one patients (35%) were clinically diagnosed with postoperative delirium. Neuron-specific enolase level ($P < 0.0001$) and the proportion of patients positive for phosphorylated neurofilament heavy subunit ($P < 0.0001$) were significantly higher in the group of patients

**Funding:** This study was supported by JSPS KAKENHI (to K. Mietani, Grant Number: 19H03749) and a Health Labour and Science Research Grant for research on chronic pain (to M. Sumitani, Grant Number: H26-Cancer-060). The funders had no role in study design, data collection and analysis, decision to publish, or preparation of the manuscript.

**Competing interests:** The Department of Pain and Palliative Medical Sciences, where M. Hasegawa-Moriyama works, is sponsored by Shionogi Co., Ltd. (Osaka, Japan) and Nippon Zoki Pharmaceutical Co., Ltd. (Osaka, Japan). This does not alter our adherence to PLOS ONE policies on sharing data and materials.

with postoperative delirium. Neuron-specific enolase level discriminated between patients with and without clinically diagnosed postoperative delirium with significantly high accuracy (area under the curve [AUC], 0.87; 95% confidence interval [CI], 0.79–0.95; $P < 0.0001$). Neuron-specific enolase level was associated with incidence of postoperative delirium independently of age (adjusted odds ratio, 8.291; 95% CI, 3.506−33.286; $P < 0.0001$). The AUC for the serum neuron-specific enolase level in detecting phosphorylated neurofilament heavy subunit was significant (AUC, 0.78; 95% CI, 0.66–0.90; $P < 0.0001$).

## Conclusion

Elevated serum neuron-specific enolase was associated with postoperative delirium independent of age as well as detection of phosphorylated neurofilament heavy subunit in serum. Serum neuron-specific enolase and phosphorylated neurofilament heavy subunit might be useful as biomarkers of postoperative delirium.

## Trial registration

University Medical Information Network (UMIN) trial ID: UMIN000010329; https://clinicaltrials.gov/.

## Introduction

Delirium is the most common complication of the central nervous system (CNS) after surgery. It usually occurs between postoperative days two and five [1]. The incidence of postoperative delirium (POD) ranges between 26 and 52% [2]. POD is associated with increased morbidity and mortality [3], prolonged intensive care unit (ICU) stay [4], and higher cost of hospital stay [5]; therefore, early detection of POD is important. However, a study of delirium assessment in the ICU showed that delirium is frequently not diagnosed when present [6]. Using delirium as diagnosed by a psychiatrist, neurologist, or geriatrician as the standard, the Confusion Assessment Method for the ICU (CAM-ICU) had a sensitivity of 64% and specificity of 82%; with sensitivity and specificity of the Intensive Care Delirium Screening Checklist (ICDSC) was 43% and specificity of 95%, respectively. The sensitivity of the ICU physician's clinical impression was only 29%. POD can lead to long-term cognitive impairment and brain atrophy even in patients who recover [7,8]. Although the causal relationship between POD and anatomical or functional CNS alterations has not been clarified, mechanism-based diagnostics and measures of POD severity in the early phases are urgently needed.

Surgical and anaesthetic stresses trigger CNS inflammations via microglial activation [9]. CNS inflammation can induce the expression of neuron-specific enolase (NSE) in neurons and S100 calcium-binding protein β (S100β) in astrocytes, which results in loss of blood-brain barrier (BBB) integrity and neurotoxicity [10]. As a consequence, neurons are structurally damaged, which may release phosphorylated neurofilament heavy subunit (pNF-H) into the cerebrospinal fluid (CSF) and peripheral blood [11]. Because pNF-H is a cytoskeletal protein specifically localized within CNS axons, it is not normally detected in the blood. Therefore, unlike NSE, which is released during neuroinflammation regardless of neuronal damage, the detection of serum pNF-H directly reflects structural neuronal damage. Previously, we reported that serum pNF-H level is directly related to clinical severity of POD [12]. In that study, serum pNF-H was detected in more than 65% of patients with POD but in less than

10% of patients without POD. Although the utility of both NSE and S100β as diagnostic markers of POD in the ICU has been investigated, their role in POD pathogenesis remains controversial [13–15]. Furthermore, the association between serum NSE and S100β levels, POD development, and detection of serum pNF-H (i.e., axonal damage) has not been thoroughly investigated.

This study aimed to explore the levels of potential CNS-derived biomarkers of POD in the acute postoperative period. First, we investigated the accuracy of NSE and S100β in POD screening; then, we evaluated the association of the levels of these markers with detection of serum pNF-H.

## Methods

### Study population

This study represents a post hoc analysis of a prospective observational study conducted at the University of Tokyo Hospital, Saitama Red Cross Hospital, and Tsukuba University Hospital [16]. Details regarding study methodology and patient characteristics may be found in our previous report. Briefly, patients were enrolled and followed up between July 23, 2013 to February 28, 2015. Serum samples from patients who participated in the initial study were collected on postoperative day 3 and stored for later testing. Patients scheduled to undergo cancer surgery under general anasthesia, irrespective of the affected organ, were eligible for inclusion. Among these we enrolled patients with an American Society of Anesthesiologists physical classification score < 4. Exclusion criteria were as follows: (1) patients with pre-existing clinically relevant cognitive dysfunction or neurological disorder; (2) patients who were prescribed tranquillizers that could influence delirium [17]; (3) patients who required neurosurgery for lesions located in the brain and/or spinal cord; and (4) patients who required cardiothoracic surgery with cardiopulmonary bypass. These neurosurgical and cardiothoracic procedures can potentially cause CNS ischemia and a consequent increase in serum pNF-H level. Therefore, patients scheduled to undergo them were excluded.

### Ethics

The local ethics committee of each institution approved the trial protocol. Written informed consent was obtained from all participants. The study was registered in the University Medical Information Network (UMIN trial ID: UMIN000010329).

### Patient assessment

Patients were assessed for delirium-associated symptoms by the attending nurses at least three times a day during regular ward rounds for the first week after surgery, as described in our previous study [16]. Suspected POD was confirmed by nurses using the CAM-ICU [18] and by investigators using ICDSC [19]. Patients diagnosed with POD within the first five days after surgery were included in the POD group; those not were included in the no POD group.

### Measurement of biomarkers

Blood samples were collected on postoperative day 3 and stored at −20˚C. Detectio of pNF-H in serum was used as a proxy for CNS axonal damage. Enzyme-linked immunosorbent assay (ELISA) was performed to measure levels of serum pNF-H (Human Phosphorylated Neurofilament H ELISA; BioVendor, Modrice, Czech Republic). The threshold for detection of pNF-H (70.5 ng/mL) was determined according to manufacturer instructions. For the measurement of NSE and S100β, bead-based multiplex assays were performed (Procarta

Immunoassay kit, human by request; Panomics Inc., Fremont, CA, USA) according to manufacturer protocol [20,21].

## Statistical analysis

Patient characteristics and biomarker levels were compared between groups using the Wilcoxon or Pearson's chi-square test. To eliminate the influence of age on serum biomarker levels, analysis of covariance (ANCOVA) was performed between groups. To identify independent biomarkers of POD, we conducted receiver operating characteristic (ROC) analyses of serum levels of NSE and S100β in participants with and without POD. NSE and S100β were analysed as dependent variables, with either dichotomous measures of postoperative delirium (i.e., presence or absence) or detection of serum pNF-H as the independent variable. Cut-off values for continuous variables were determined using the Youden index [22,23]. Subsequently, multiple logistic regression was performed with direct entry of variance using a model based on the log-transformed concentrations of the potential candidate variables. Analyses were performed using JMP Pro 15 software (SAS Institute, Cary, NC, USA) and SPSS software version 22 (IBM Corp, Armonk, NY, USA). $P \leq 0.05$ was considered significant.

## Results

### Patient characteristics

A total of 119 patients who underwent elective cancer surgery under general anesthesia were eligible for study inclusion, and 117 patients were analyzed (Fig 1). The baseline characteristics of the 117 patients with and without postoperative delirium are presented in Table 1. Age was significantly higher in the patients with POD ($P < 0.0001$).

### Association between POD and CNS-derived biomarkers

Forty-one patients (35.0%) were clinically diagnosed with POD (Table 1). The proportion of patients whowere positive for pNF-H (Table 1) and serum levels of pNF-H (coefficient of variation [CV]: no POD, 161.6; POD, 533.1), NSE (CV: no POD, 31.5; POD, 69.7), and S100β (CV: no POD, 94.2; POD, 119.0) were significantly higher in the POD group (Fig 2). After performing ANCOVA, using age as a covariate (S1 Table), the levels of pNF-H, NSE, and S100 β remained significantly higher in the POD group.

Using a cut-off value of 201.2 ng/mL, the area under the curve (AUC) for serum NSE level in predicting delirium was 0.87 (sensitivity, 0.76; specificity, 1.00; 95% confidence interval [CI], 0.79–0.95; Fig 3). In contrast, serum S100β level exhibited lower correlation using a cut-off value of 305 pg/mL (AUC, 0.74; sensitivity, 0.78; specificity, 0.61; 95% CI, 0.64–0.83). Multivariate logistic regression analysis showed that age (adjusted odds ratio [OR], 1.080; 95% CI, 1.8–1.172; $P = 0.0269$) and NSE (adjusted OR, 8.291; 95% Cl, 3.506–33.286; $P < 0.0001$; Table 2) were associated with POD.

### Association between the detection of pNF-H and CNS-derived biomarkers

The association between the detection of pNF-H and serum levels of NSE and S100β is shown in Table 3. Serum NSE level was significantly higher in patients with detectable pNF-H. Although serum S100β level was also higher in the patients with detectable pNF-H, the difference was not significant.

Using a cut-off value of 201.2 ng/mL, the AUC for serum NSE level in predicting detection of pNF-H was 0.78 (sensitivity, 0.73; specificity, 0.90; 95% CI, 0.66–0.90; Fig 4). In contrast,

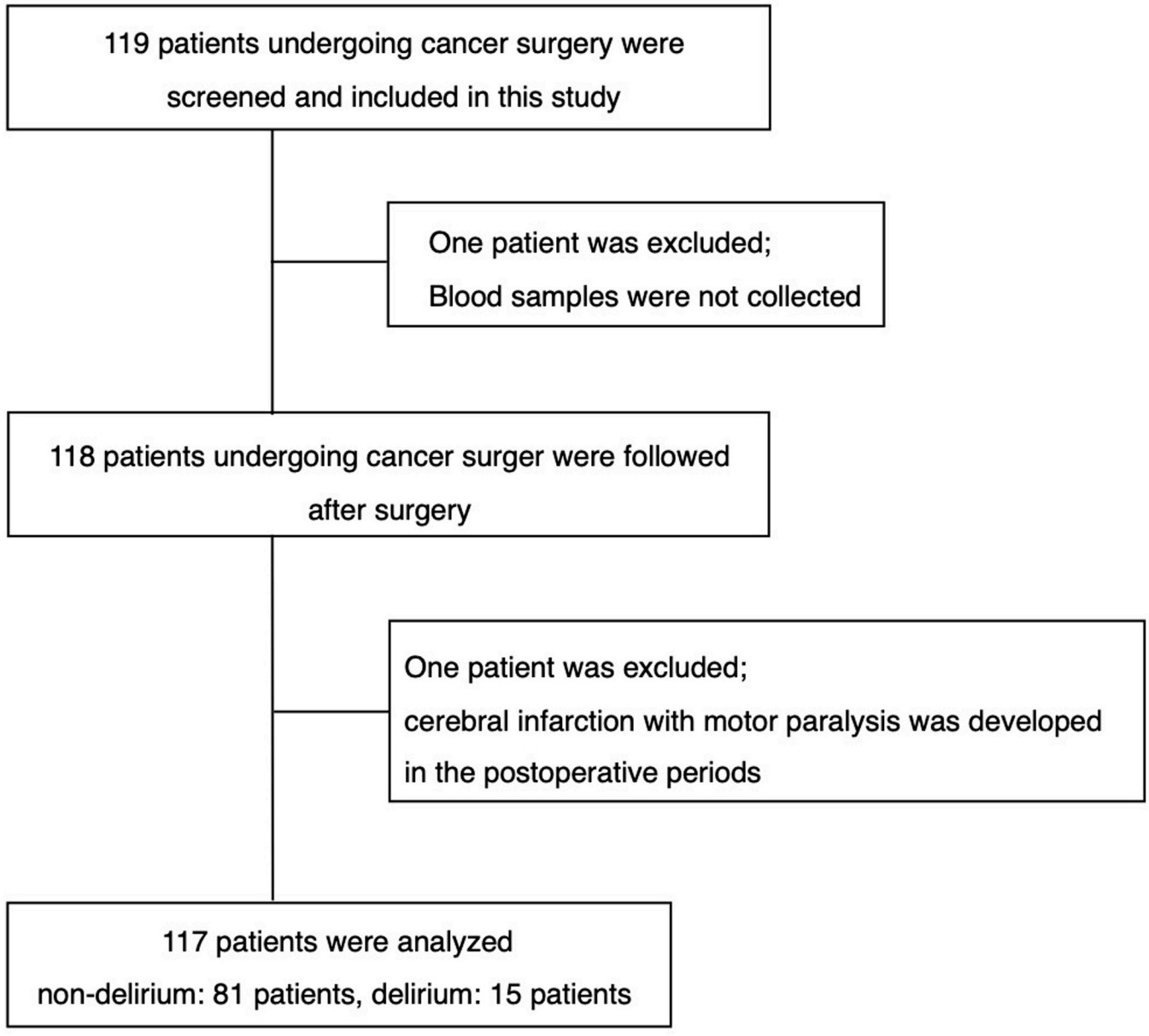

**Fig 1. Study flowchart.**

serum S100β level exhibited a lower association (AUC, 0.61; sensitivity, 0.53; specificity, 0.69; 95% CI, 0.49–0.72) using a cut-off value of 502.0 pg/mL/.

## Discussion

This study examined the clinical usefulness of serum CNS-derived biomarkers in diagnosing POD. NSE level was significantly higher in POD patients (Fig 2), and significantly associated with POD independent of age (Table 2). Furthermore, NSE level showed high accuracy for diagnosing POD (Fig 3). Because NSE level was associated with POD independent of age, this marker should be useful in elderly patients. Although both NSE and S100β in serum indicates a loss of BBB integrity, only NSE level was significantly associated with detection of serum

**Table 1. Characteristics of patients grouped according to development of postoperative delirium.**

|  | POD[a] (n = 41) | no POD[a] (n = 76) | P value |
|---|---|---|---|
| Age; y | 78 (73–81) | 67 (58–74) | < 0.0001 |
| Gender; male | 24 (58.5%) | 40 (52.6%) | 0.54 |
| BMI[b]; kg/m$^2$ | 22.8 (20.2–24.1) | 21.8 (19.8–24.1) | 0.32 |
| pNF-H[c] positive (≥ 70.5 pg/mL) | 23 (56.1%) | 7 (9.2%) | < 0.0001 |

Values are presented as numbers (proportion) or medians (interquartile range).

[a]POD, postoperative delirium;

[b]BMI, body mass index;

[c]pNF-H, phosphorylated neurofilament heavy subunit.

pNF-H in our study (Table 3, Fig 4) [10]. S100β is mainly located in astrocytes, whose end-foot processes interacts with the BBB [10]. The reason that serum NSE increases without an accompanying increase in S100β is unclear. In our previous study [16]. we reported P-selection, which is expressed on endothelial cells and involved in recruitment of circulating leukocytes [24], is independently associated with detection of serum pNF-H. Taken together, compared with astrocytes, neurons might be structurally vulnerable against a loss of BBB integrity, However, anticancer agents induce oxidative stress, DNA damage, and dysregulation of neuronal repair processes [25] and serum pNF-H increases in a cumulative dose-dependent manner in breast cancer patients undergoing chemotherapy [26]. Considering that some patients in our study were undergoing chemotherapy at the time of blood sampling, the toxic effect of these drugs on neurons may have affected our results. To clarify the influence of chemotherapy-induced neurotoxicity in future studies, baseline preoperative biomarker levels should be further examined in addition to postoperative day 3 levels.

Serum NSE level was highly associated with POD and detection of pNF-H, which is proxy of for CNS axonal damage, suggesting that NSE can reflect both the onset and severity of POD. Our ROC analysis showed that the serum NSE cut-off values for the diagnosis of POD and detection of pNF-H were same (201.2 ng/mL for both; Figs 3 and 4). These results suggest that POD is associated with the development of inflammation-induced neuronal damage.

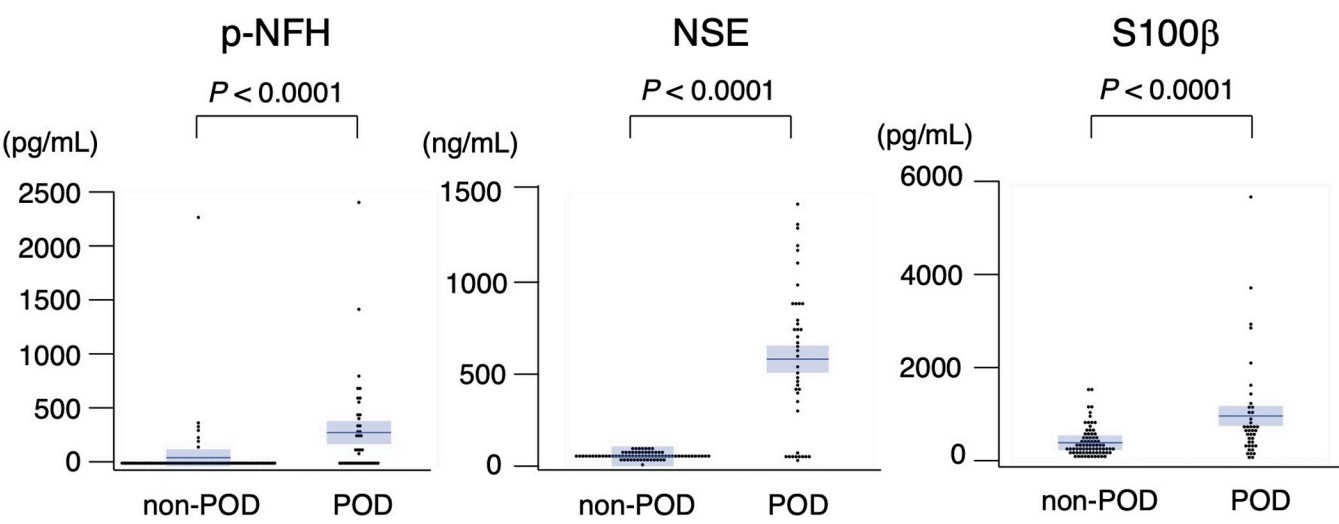

**Fig 2. Comparison of serum biomarker levels in patients with and without postoperative delirium.**

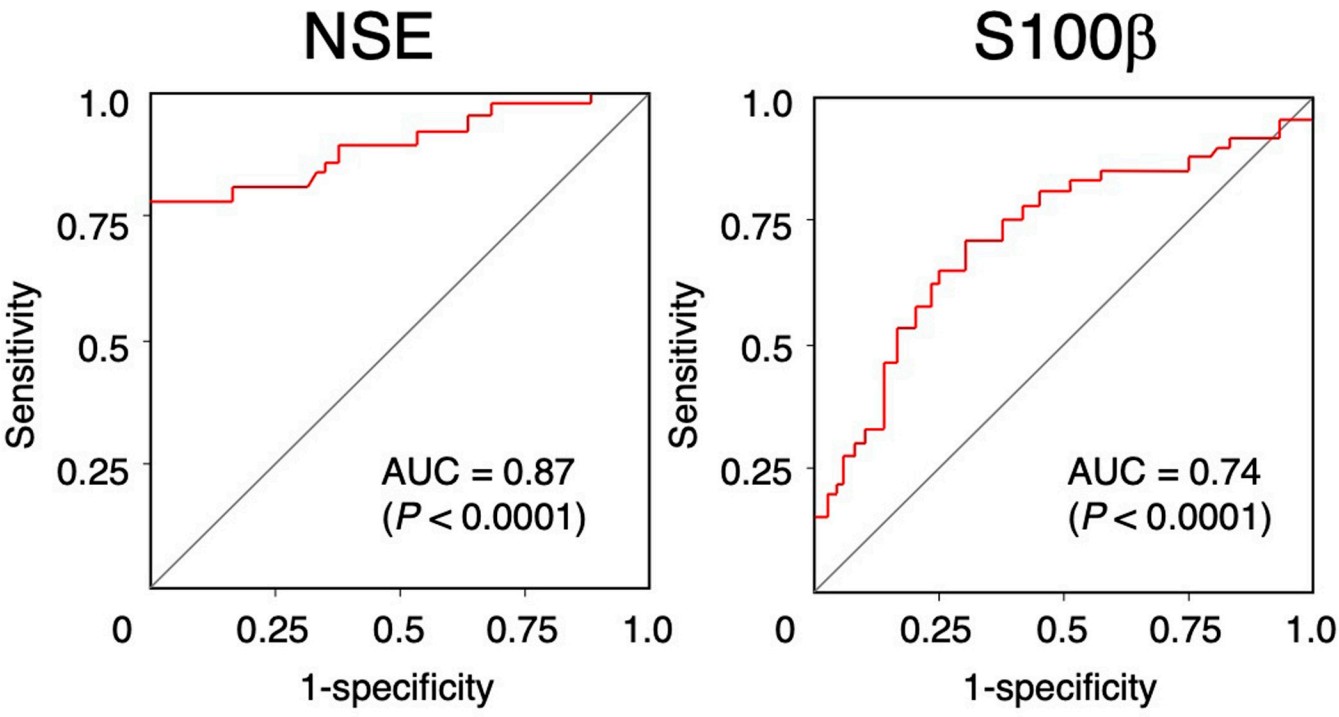

**Fig 3. Receiver operating characteristic curves analysis of serum biomarker levels for the diagnosis of postoperative delirium.** NSE, neuron-specific enolase; S100β, S100 calcium-binding protein β.

**Table 2. Logistic regression analysis for prediction of postoperative delirium.**

|  | Unadjusted OR[a] (95% CI[b]) | P value | Adjusted OR[a] (95% CI[b]) | P value |
|---|---|---|---|---|
| age[c]; y | 1.139 (1.081−1.215) | < 0.0001 | 1.080 (1.008−1.172) | 0.0269 |
| log (pNF-H[d]) | 1.533 (1.292−1.819) | < 0.0001 | 1.105 (0.761−1.527) | 0.567 |
| log (NSE[e]) | 9.972 (4.667−34.648) | < 0.0001 | 8.291 (3.506−33.286) | < 0.0001 |
| log (S100β[f]) | 2.042 (1.371−3.210) | 0.0002 | 1.580 (0.842−2.997) | 0.1466 |

[a]odds ratio;

[b]confidential interval.

[c]OR for age per each additional year of age.

[d]phosphorylated neurofilament heavy subunit;

[e]neuron-specific enolase;

[f]S100 calcium-binding protein β.

**Table 3. Association between detection of phosphorylated neurofilament heavy subunit and candidate biomarker levels.**

|  | pNF-H[a] positive (n = 30) | pNF-H[a] negative (n = 87) | P value |
|---|---|---|---|
| NSE[b]; ng/mL | 529.010 (758.925−763.164) | 54.164 (45.144−63.656) | < 0.0001 |
| S100 β[c]; pg/mL | 528.8 (211.9−706.8) | 304.7 (143.7−616.7) | 0.0789 |

Values are presented as medians (interquartile range).

[a]phosphorylated neurofilament heavy subunit;

[b]NSE: Neuron-specific enolase;

[c]S100 calcium-binding protein β.

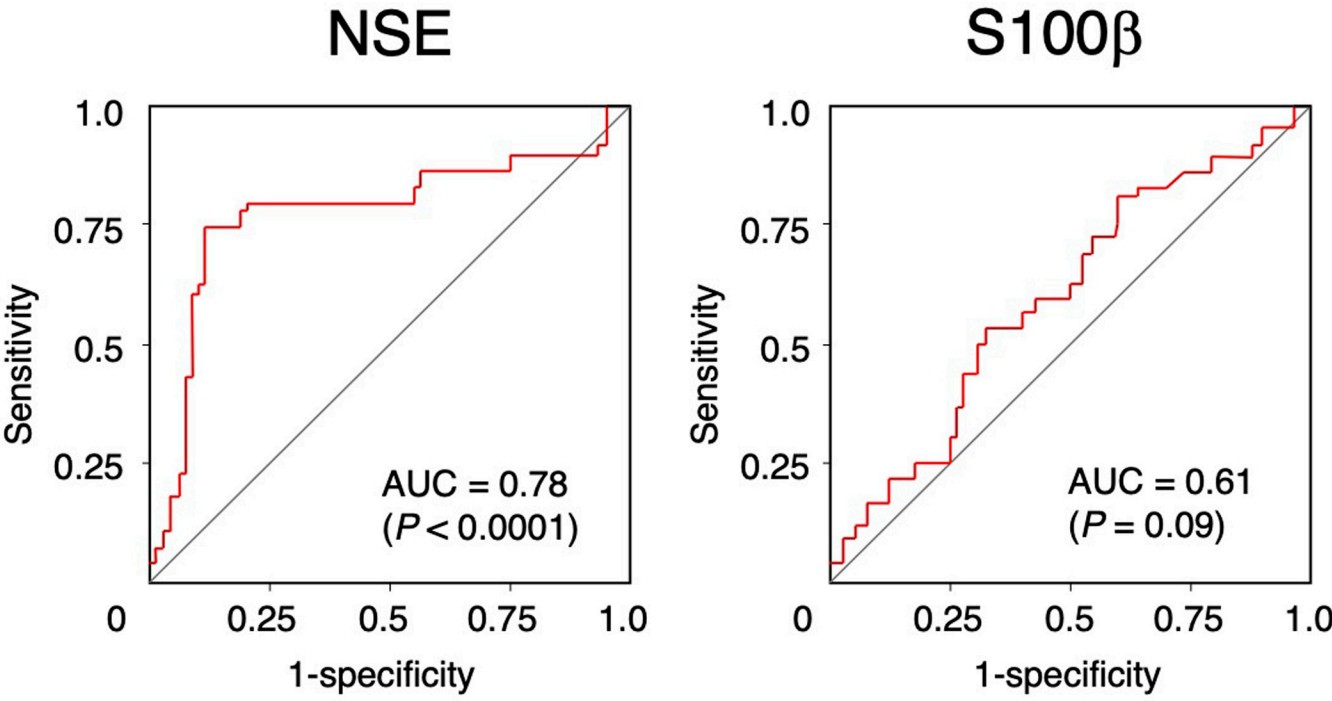

**Fig 4. Receiver operating characteristic curves analysis of serum biomarker levels for detection of phosphorylated neurofilament heavy subunit.** NSE, neuron-specific enolase; S100β, S100 calcium-binding protein β.

This study has several limitations. First, the biological half-lives of serum NSE and S100β (approximately 48 hours each) are shorter than that the pNF-H half-life (approximately 96 hours) [27]. Therefore, we cannot exclude the possibility that a shorter time window may be required to follow the severity of delirium by NSE than pNF-H. Second, this study was a post hoc analysis of a previous prospective observational study [16]. Therefore, the causal relationship between the onset and severity of POD and changes in biomarkers over time should be further investigated in a prospective manner. Third, we excluded patients with clinically relevant pre-existing cognitive dysfunction or neurological disorders because serum pNF-H level increases in certain CNS disorders, such as spinal cord injury, Alzheimer's disease, and febrile convulsions [28–30], and such dysfunction or disorders may share symptoms and signs with POD. Finally, although NSE was significantly associated with POD independent of age, multiple logistic regression cannot completely control for each variable. Therefore, further investigation of the association between age and levels of CNS-derived biomarkers is warranted.

## Conclusion

NSE is an accurate diagnostic biomarker for POD that is highly associated with detection of pNF-H, a proxy for CNS axonal damage. These findings indicate that the timing of neuronal structural changes might be consistent with both neuronal changes and the onset of POD.

Diagnostic accuracy of POD might be increased with early monitoring of serum NSE level in combination with clinical assessment.

## Supporting information

**S1 Table. Comparison of biomarkers using analysis of covariance with age as a covariate.** (XLSX)

## Acknowledgments

We thank the patients who participated in this study and the clinical and nursing staff who provided their care. In addition, we thank Edanz (https://jp.edanz.com/ac) for editing a draft of this manuscript.

## Author Contributions

**Conceptualization:** Masahiko Sumitani.

**Data curation:** Maiko Hasegawa-Moriyama.

**Formal analysis:** Maiko Hasegawa-Moriyama.

**Investigation:** Kazuhito Mietani, Reo Inoue, Toru Ogata, Nobutake Shimojo, Makoto Kurano.

**Writing – original draft:** Maiko Hasegawa-Moriyama.

**Writing – review & editing:** Masahiko Sumitani, Kanji Uchida.

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
