## [Decision Letter · Decision Letter 0]

30 Sep 2021

PONE-D-21-24158Elevation of neuron-specific enolase predicts postoperative delirium accompanying axonal damage: a prospective observational studyPLOS ONE

Dear Dr. Sumitani,

Thank you for submitting your manuscript to PLOS ONE. After careful consideration, we feel that it has merit but does not fully meet PLOS ONE’s publication criteria as it currently stands. Therefore, we invite you to submit a revised version of the manuscript that addresses the points raised during the review process.

We look forward to receiving your revised manuscript.

Kind regards,

Aleksandar R. Zivkovic

Academic Editor

PLOS ONE

Journal Requirements:

This study was supported by JSPS KAKENHI (to K. Mietani, Grant Number: 19H03749) and a Health Labour and Science Research Grant for research on chronic pain (to M. Sumitani, Grant Number: H26-Cancer-060).

3. Thank you for stating the following in the Acknowledgments/ Funding Section of your manuscript: 

This study was supported by JSPS KAKENHI (to K. Mietani, Grant Number: 19H03749) and a Health Labour and Science Research Grant for research on chronic pain (to M. Sumitani, Grant Number: H26-Cancer-060).

This study was supported by JSPS KAKENHI (to K. Mietani, Grant Number: 19H03749) and a Health Labour and Science Research Grant for research on chronic pain (to M. Sumitani, Grant Number: H26-Cancer-060).

The Department of Pain and Palliative Medical Sciences, where M. Hasegawa-Moriyama works, is sponsored by Shionogi Co., Ltd. (Osaka, Japan) and Nippon Zoki Pharmaceutical Co., Ltd. (Osaka, Japan).

Reviewers' comments:

Reviewer #1: This study evaluated plasma levels of (3) proteins in patients that underwent surgery for cancer-associated conditions. The proteins of interest (phospho-neurofilament, neuron-specific enolase, and S100) have all been previously associated with delirium, so this study does not particularly add to the literature since they used a very targeted approach, rather than an unbiased, untargeted method to look at different groups of proteins. Suggestions are made to improve the quality of the manuscript.

-Please report the median with the interquartiles and 95% CI for all proteins reported, rather than only the mean +/- stdev. Graphical representation of the distribution of plasma levels comparing control vs delirium groups (including individual data points for each patient sample) would be beneficial to the reader.

-The % coefficient of variation should be reported for all assays performed, distinguishing between technical replicates and biological replicates.

-Please clearly state in the methods section the timeframe for evaluation of biomarker associations with delirium (i.e., delirium development anytime within the week, before or after plasma collection).

-Since the patient population that developed postop delirium was significantly older than the non-delirium group, age should be considered within the analysis.

-Please include an analysis comparing biomarker levels (in plasma collected on day 3) to the delirium severity and/or incidence on that day.

-Some of the statements go too far in their conclusion given the very limited biochemical analysis performed in this study (ex- pg. 12, lines 6-8). Please revise throughout.

Reviewer #2: General comments:

My biggest criticism of this study is that the authors claim, "to explore the changes in potential CNS-derived biomarkers…" yet I did not see any evidence that change has been studied. The best I can tell, this is a cross-sectional study of associations between POD and a handful of biomarkers. The discussion is more measured, but causality still creeps into the discussion.

The authors place a high level of faith in regression models to mitigate the difference in age between the POD positive and negative groups. While it is true that type III effects are calculated after account for the variability accounted for from other covariates, regression may not entirely remove the effect of covariates. For instance, if the effect of age is nonlinear or interacts with the variable of interest, controlling for the main effect will not fully remove the differences between groups.

Otherwise, this seems to be a fairly standard regression analysis. I have questions about how this study fits into the greater body of research since it sounds like associations between POD and biomarkers have been studied, I don't think this addresses pathogenesis. Hopefully other reviewers can address this.

I have other, smaller criticisms that are included in my specific comments below.

Specific comments (my page numbers are based on the PDF build from PLOS since I couldn't find page numbers on every page; it's helpful for reviewers to have page numbers and, better yet, line numbers that are continuous throughout the paper):

1. (abstract, lines 9-10) I think the abstract should at least mention how patients were sampled and, preferably, something about the analyses performed.

2. (p.12, line 13) Something like "Mietani et al" is needed here.

3. (p.12) Where were patients ascertained?

4. (p.13, lines 18-19) Please provide a citation for ROC analysis.

5. (p.14, line 3) Please provide a citation for the Youden index.

6. (p.14, lines 4-7) It's unclear why both bivariate screening (the p<0.1 part) and stepwise selection (lines 166-7) are used. Though, the larger problem is, with the sample sizes you have, stepwise variable selection procedures usually do a poor job of finding the most appropriate model (e.g., https://doi.org/10.1002/sim.3943). Stepwise procedures and any p-value based selection have quite a bit of evidence suggesting that they are suboptimal at selecting the appropriate variables. For a decent summary, see the link above. Generally, it's better to select based on more robust criteria, especially measures which assess the fit of the model, such as BIC, or, better yet, a shrinkage-based estimator such as lasso or lars.

7. (p.16, lines 5-7; p.17 lines 7-11) The sensitivity and specificity estimates should have 95% confidence intervals.

8. (Table 2) What are the units for the OR for age? For instance, is the OR for age per each additional year of age?

9. (Table 2) I suggest including the results for all variables considered instead of only those that are included in the final model.

10. (p.16 line 16; Table 3; p.17, line 10; p.18, line 4, …) In this context, I believe the term "association" would be preferred over "correlation".

Reviewer #3: Assistant professor Mietami and colleagues present results from a prospective observational trial examining the ability of neuromarkers to predict the occurrence of postoperative delirium, as well as the ability of the neuroinflammatory markers S100b and neuron-specific enolase to predict axonal damage assessed by neurofilament heavy subunit.

The article is well written, the methods are sound, and the statistical analyses appear correct - I would like to congratulate the authors for their work.

I have some suggestions for the authors:

1) I would suggest writing that the results originate from a 'prospective observational trial', which I believe is the case. It would facilitate reading, if this was defined in the methods section without the need to search for the reference of the preciously published paper.

2) It is not clear to me, whether the analyses were pre-planned, or whether the present analyses were conducted post hoc. This has implications for the interpretation and applicability of the results, and I think it should be written in the methods section. Further, I believe that it should be mentioned in the limitations section that the results should be considered hypothesis generating, and that the suggested NSE cut-offs should be validated externally in future prospective cohorts.

3) The study included 117 patients. It is unclear, whether the included patients were consecutive or not. If possible, a consort diagram showing in- and exclusions should be provided. As a minimum it should be defined more clearly, how patients were selected i.e. is there a risk of selection bias.

4) You present biomarker levels as picograms per milliliter. In most studies the presented unit of NSE is nanograms per milliliter. Can you provide a reference for the precision of your NSE assay? If the assay has a high enough precision to assess picograms per milliliter it is fine, but if the precision is lower, I would suggest presenting biomarkers as nanograms per milliliter in stead.

5) The NSE values you present seem very high compared to other studies. In resuscitated cardiac arrest, patients usually have NSE values from 20 - 200 nanograms per milliliter, and any value above 100 ng/mL is a strong predictor of very poor neurologic outcome or death (Stammet et al, JACC 2015). In patients undergoing cardiac surgery with cardiopulmonary bypass, NSE values usually range from 4 - 20 ng/mL. Can you comment on, whether the high values are due to the application of a different assay, or any other explanation? Or is there an error regarding the units you present?

6. PLOS authors have the option to publish the peer review history of their article (what does this mean?). If published, this will include your full peer review and any attached files.

Reviewer #1: No

Reviewer #2: No

Reviewer #3: **Yes: **Sebastian Wiberg

---

## [Author Response · Author response to Decision Letter 0]

13 Oct 2021

Dr. Aleksandar R. Zivkovic

Academic Editor

PLOS ONE

PONE-D-21-24158

Elevated neuron-specific enolase level is associated with postoperative delirium and detection of phosphorylated neurofilament heavy subunit: A prospective observational study

Dear Dr. Zivkovic

We would like to thank you and the reviewers for assessing our manuscript and providing comments, which have helped us to improve the quality and clarity of our manuscript. Below, we have addressed each of the reviewer’s comments in blue and have highlighted all changes made to the manuscript in red. 

We rewrote our revised manuscript to meet the PLOS ONE style formatting.

This study was supported by JSPS KAKENHI (to K. Mietani, Grant Number: 19H03749) and a Health Labour and Science Research Grant for research on chronic pain (to M. Sumitani, Grant Number: H26-Cancer-060).

We added the statement of financial disclosure and its role to the revised cover letter.

3. Thank you for stating the following in the Acknowledgments/ Funding Section of your manuscript:

This study was supported by JSPS KAKENHI (to K. Mietani, Grant Number: 19H03749) and a Health Labour and Science Research Grant for research on chronic pain (to M. Sumitani, Grant Number: H26-Cancer-060).

This study was supported by JSPS KAKENHI (to K. Mietani, Grant Number: 19H03749) and a Health Labour and Science Research Grant for research on chronic pain (to M. Sumitani, Grant Number: H26-Cancer-060).

We removed the above funding-related text from the manuscript. 

We have no funding status updates to provide.

The Department of Pain and Palliative Medical Sciences, where M. Hasegawa-Moriyama works, is sponsored by Shionogi Co., Ltd. (Osaka, Japan) and Nippon Zoki Pharmaceutical Co., Ltd. (Osaka, Japan).

We removed the competing interests statement from our manuscript and added it to the revised cover letter. We declared that this does not alter our adherence to PLOS ONE policies on sharing data and materials.

Reviewers' comments:

Reviewer #1: This study evaluated plasma levels of (3) proteins in patients that underwent surgery for cancer-associated conditions. The proteins of interest (phospho-neurofilament, neuron-specific enolase, and S100) have all been previously associated with delirium, so this study does not particularly add to the literature since they used a very targeted approach, rather than an unbiased, untargeted method to look at different groups of proteins. Suggestions are made to improve the quality of the manuscript.

-Please report the median with the interquartiles and 95% CI for all proteins reported, rather than only the mean +/- stdev. Graphical representation of the distribution of plasma levels comparing control vs delirium groups (including individual data points for each patient sample) would be beneficial to the reader.

Values are presented as medians (interquartile range) in our new Table 1. In addition, we added a graph indicating the serum levels of CNS-derived proteins of pNF-H, NSE, and S100� in the POD and no POD groups to our new Figure 2. Individual data points for each patient sample have been added as suggested. 

-The % coefficient of variation should be reported for all assays performed, distinguishing between technical replicates and biological replicates.

We added the % coefficient of variation of pNF-H, NSE, and S100� on Page 10, line 18.

The proportion of patients whowere positive for pNF-H (Table 1) and serum levels of pNF-H (coefficient of variation [CV]: no POD, 161.6; POD, 533.1), NSE (CV: no POD, 31.5; POD, 69.7), and S100� (CV: no POD, 94.2; POD, 119.0) were significantly higher in the POD group (Fig 2).

-Please clearly state in the methods section the timeframe for evaluation of biomarker associations with delirium (i.e., delirium development anytime within the week, before or after plasma collection).

As suggested, we added the following sentence to Page 8, line 5:

Patients diagnosed with POD within the first 5 days after surgery were included in the POD group; those not were included in the no POD group.

-Since the patient population that developed postop delirium was significantly older than the non-delirium group, age should be considered within the analysis.

We included age as a covariate for the analysis in our new Supplementary Table 1. 

-Please include an analysis comparing biomarker levels (in plasma collected on day 3) to the delirium severity and/or incidence on that day.

Patients who developed delirium within 5 days after surgery were included in the POD group. Because patient records are preserved only for 5 years, we could not re-evaluate the relationship between onset and day of sumple collection Therefore, we added the following sentences to the limitations section of the discussion on Page 15, line 4

Second, this study was a post hoc analysis of a previous prospective observational study [16]. Therefore, the causal relationship between the onset and severity of POD and changes in biomarkers over time should be further investigated in a prospective manner.

-Some of the statements go too far in their conclusion given the very limited biochemical analysis performed in this study (ex- pg. 12, lines 6-8). Please revise throughout.

The relevant descriptions in the discussion and conclusion were rewritten throughout the manuscript.

Reviewer #2: General comments:

My biggest criticism of this study is that the authors claim, "to explore the changes in potential CNS-derived biomarkers…" yet I did not see any evidence that change has been studied. The best I can tell, this is a cross-sectional study of associations between POD and a handful of biomarkers. The discussion is more measured, but causality still creeps into the discussion.

The authors place a high level of faith in regression models to mitigate the difference in age between the POD positive and negative groups. While it is true that type III effects are calculated after account for the variability accounted for from other covariates, regression may not entirely remove the effect of covariates. For instance, if the effect of age is nonlinear or interacts with the variable of interest, controlling for the main effect will not fully remove the differences between groups.

Page 11, line 1

We performed analysis of covariance with regarding age as a covariate in Supplementary table 1.

After performing ANCOVA using age as a covariate (Supplementary table 1), the level of pNF-H, NSE, and S100� remained significantly higher in the POD group.

In addition, the following sentence was added to the limitations:

Page 15, line 11

Finally, although NSE was statistically associated with POD independent of age, multiple logistic regression cannot completely control for each variable. Therefore, further investigation of the association between age and levels of CNS-derived biomarkers is warranted.

Otherwise, this seems to be a fairly standard regression analysis. I have questions about how this study fits into the greater body of research since it sounds like associations between POD and biomarkers have been studied, I don't think this addresses pathogenesis. Hopefully other reviewers can address this.

As suggested by both reviewers 1 and 2, the possible overstatements concerning biological changes and pathogenesis were rewritten.

I have other, smaller criticisms that are included in my specific comments below.

Specific comments (my page numbers are based on the PDF build from PLOS since I couldn't find page numbers on every page; it's helpful for reviewers to have page numbers and, better yet, line numbers that are continuous throughout the paper):

We added page numbers to the revised manuscript.

1. (abstract, lines 9-10) I think the abstract should at least mention how patients were sampled and, preferably, something about the analyses performed.

We added the following sentences to the Methods of abstract:

Page 3, line 9

This study represents a post hoc analysis of 117 patients who participated in a prospective observational study of postoperative delirium in patients undergoing cancer surgery. Patients were clinically assessed for development of postoperative delirium within the first five days of surgery.

2. (p.12, line 13) Something like "Mietani et al" is needed here.

As suggested, we added “in our previous study” in the following sentence:

Page 13, line 16

In our previous study [16], we reported that P-selection, which is expressed on endothelial cells is involved in recruitment of circulating leukocytes [24], is independently associated with detection of serum pNF-H.

3. (p.12) Where were patients ascertained?

Patients were assessed postoperatively by the nursing staff during the regular ward rounds in the first week after surgery as indicated in the method section. Therefore, the patients were assessed in the ICU initially and then the surgical floor after transfer from the ICU.

4. (p.13, lines 18-19) Please provide a citation for ROC analysis.

The following literature was cited on page 9, line 2.

22. Lin X, Tang J, Liu C, Li X, Cao X, Wang B et al. Cerebrospinal fluid cholinergic biomarkers are associated with postoperative delirium in elderly patients undergoing Total hip/knee replacement: a prospective cohort study. BMC Anesthesiol 2020; 20: 246.

5. (p.14, line 3) Please provide a citation for the Youden index.

The following literature was cited on page 9, line 2.

23. Bantis LE, Nakas CT, Reiser B. Construction of confidence regions in the ROC space after the estimation of the optimal Youden index-based cut-off point. Biom J 2005; 47: 458–472. 

6. (p.14, lines 4-7) It's unclear why both bivariate screening (the p<0.1 part) and stepwise selection (lines 166-7) are used. Though, the larger problem is, with the sample sizes you have, stepwise variable selection procedures usually do a poor job of finding the most appropriate model (e.g., https://doi.org/10.1002/sim.3943). Stepwise procedures and any p-value based selection have quite a bit of evidence suggesting that they are suboptimal at selecting the appropriate variables. For a decent summary, see the link above. Generally, it's better to select based on more robust criteria, especially measures which assess the fit of the model, such as BIC, or, better yet, a shrinkage-based estimator such as lasso or lars.

Instead of stepwise variable selection procedures, we used direct entry in our new Table 3.

Page 9, line 2

Subsequently, multiple logistic regression was performed with direct entry of variance using a model based on the log-transformed concentrations of the potential candidate variables.

7. (p.16, lines 5-7; p.17 lines 7-11) The sensitivity and specificity estimates should have 95% confidence intervals.

We have added 95% confidence intervals to the data in the place you have mentioned.

8. (Table 2) What are the units for the OR for age? For instance, is the OR for age per each additional year of age?

Thank you for your comments. The OR for age is per each additional year of age, which has been added to our new Table 2.

9. (Table 2) I suggest including the results for all variables considered instead of only those that are included in the final model.

We have added pNF-H and S100� data to our new Table 2.

10. (p.16 line 16; Table 3; p.17, line 10; p.18, line 4, …) In this context, I believe the term "association" would be preferred over "correlation".

We changed the term “correlation” to “association” throughout the manuscript.

Reviewer #3: Assistant professor Mietami and colleagues present results from a prospective observational trial examining the ability of neuromarkers to predict the occurrence of postoperative delirium, as well as the ability of the neuroinflammatory markers S100b and neuron-specific enolase to predict axonal damage assessed by neurofilament heavy subunit.

The article is well written, the methods are sound, and the statistical analyses appear correct - I would like to congratulate the authors for their work.

I have some suggestions for the authors:

1) I would suggest writing that the results originate from a 'prospective observational trial', which I believe is the case. It would facilitate reading, if this was defined in the methods section without the need to search for the reference of the preciously published paper.

The following sentence was rewritten as suggested.

Page 7, line 3

This study represents a post hoc analysis of a prospective observational study conducted at the University of Tokyo Hospital, Saitama Red Cross Hospital, and Tsukuba University Hospital [16].

2) It is not clear to me, whether the analyses were pre-planned, or whether the present analyses were conducted post hoc. This has implications for the interpretation and applicability of the results, and I think it should be written in the methods section. Further, I believe that it should be mentioned in the limitations section that the results should be considered hypothesis generating, and that the suggested NSE cut-offs should be validated externally in future prospective cohorts.

As noted above in our response to your first suggestion, this was a post hoc analysis of previously obtained data, which is now specified early in the methods section.

Further, I believe that it should be mentioned in the limitations section that the results should be considered hypothesis generating, and that the suggested NSE cut-offs should be validated externally in future prospective cohorts.

Page 15, line 4

Second, this study was a post hoc analysis of a previous prospective observational study [16]. Therefore, the causal relationship between the onset and severity of POD and changes in biomarkers over time should be further investigated in a prospective manner.

Page 15, line 11

Finally, although NSE was statistically associated with POD independent of age, multiple logistic regression cannot completely control for each variable. Therefore, further investigation of the association between age and levels of CNS-derived biomarkers is warranted.

3) The study included 117 patients. It is unclear, whether the included patients were consecutive or not. If possible, a consort diagram showing in- and exclusions should be provided. As a minimum it should be defined more clearly, how patients were selected i.e. is there a risk of selection bias.

We show a study flow chart in our new Figure 1.

4) You present biomarker levels as picograms per milliliter. In most studies the presented unit of NSE is nanograms per milliliter. Can you provide a reference for the precision of your NSE assay? If the assay has a high enough precision to assess picograms per milliliter it is fine, but if the precision is lower, I would suggest presenting biomarkers as nanograms per milliliter in stead.

NSE levels are as nanogram/mL as suggested. A multiplex immunoassay system was used to measure biomarker levels, This system detects protein with higher sensitivity than conventional ELISA. 

We added the following references 20 and 21 to the Methods.

20. Menzenbach J, Frede S, Petras J, Guttenthaler V, Kirfel A, Neumann C et al. Perioperative Vascular Biomarker Profiling in Elective Surgery Patients Developing Postoperative Delirium: A Prospective Cohort Study. Biomedicines. 202; 9: 553. 

21. Zhang Y., Birru R., Di Y.P. Analysis of Clinical and Biological Samples Using Microsphere-Based Multiplexing Luminex System. Mol. Toxicol. Protoc. 2014; 1105: 43–57. 

5) The NSE values you present seem very high compared to other studies. In resuscitated cardiac arrest, patients usually have NSE values from 20 - 200 nanograms per milliliter, and any value above 100 ng/mL is a strong predictor of very poor neurologic outcome or death (Stammet et al, JACC 2015). In patients undergoing cardiac surgery with cardiopulmonary bypass, NSE values usually range from 4 - 20 ng/mL. Can you comment on, whether the high values are due to the application of a different assay, or any other explanation? Or is there an error regarding the units you present?

Please refer to our comments above. 

Menzenbach J et al. addresses the limitations of the multiplex system in their report as follows: the use of multiplex arrays that allow for the measurement of a substantial number of biomarkers at once, while using only small serum sample sizes, could also have resulted in less accurate detection of some proteins over others.

---

## [Editor Report · Decision Letter 1]

15 Oct 2021

Elevated neuron-specific enolase  level is associated with  postoperative delirium and detection of phosphorylated neurofilament heavy subunit: A prospective observational study

PONE-D-21-24158R1

Dear Dr. Sumitani,

We’re pleased to inform you that your manuscript has been judged scientifically suitable for publication and will be formally accepted for publication once it meets all outstanding technical requirements.

Kind regards,

Aleksandar R. Zivkovic

Academic Editor

PLOS ONE

---

## [Editor Report · Acceptance letter]

10 Nov 2021

PONE-D-21-24158R1 

Elevated neuron-specific enolase level is associated with postoperative delirium and detection of phosphorylated neurofilament heavy subunit: A prospective observational study 

Dear Dr. Sumitani:

I'm pleased to inform you that your manuscript has been deemed suitable for publication in PLOS ONE. Congratulations! Your manuscript is now with our production department. 

Kind regards, 

on behalf of

Dr. Aleksandar R. Zivkovic 

Academic Editor

PLOS ONE